# Differentiable Analog Quantum Computing for Optimization and Control

**Jiaqi Leng**[†,1,3]   **Yuxiang Peng**[†,1,2]   **Yi-Ling Qiao**[†,2,4]   **Ming C. Lin**[2,4]   **Xiaodi Wu**[1,2]

[1]Joint Center for Quantum Information and Computer Science, University of Maryland
[2]Department of Computer Science, University of Maryland
[3]Department of Mathematics, University of Maryland
[4]Center for Machine Learning, University of Maryland
[†]Equal Contribution

## Abstract

We formulate the first differentiable analog quantum computing framework with specific parameterization design at the analog signal (pulse) level to better exploit near-term quantum devices via variational methods. We further propose a scalable approach to estimate the gradients of quantum dynamics using a forward pass with Monte Carlo sampling, which leads to a quantum stochastic gradient descent algorithm for scalable gradient-based training in our framework. Applying our framework to quantum optimization and control, we observe a significant advantage of differentiable analog quantum computing against SOTAs based on parameterized digital quantum circuits by *orders of magnitude*.

## 1 Introduction

Quantum computing has promised unprecedented improvement in our computational ability to tackle classically intractable problems. Despite of the rapid development of quantum hardware [2, 66], near-term quantum computers are still likely to have very restricted hardware resources, where the limited number of "qubits" and non-negligible machine noises would impede the implementation of large-scale quantum algorithms. Recent research results in both computer science [60] and physics [43] suggest a promising approach of designing resource-efficient *Noisy Intermediate-Scale Quantum (NISQ)* [51] applications by breaking quantum circuit abstractions and directly designing applications at the pulse-level control of quantum machines.[1] The benefits of this analog-oriented approach have been witnessed in the history of classical analog computing that predates digital computing due to relaxed hardware requirement and plays an important role in domain applications such as simulation.

One leading algorithmic paradigm on NISQ computers is the *Variational Quantum Algorithm (VQA)* with a few prominent examples like the Variational Quantum Eigensolver (VQE) [50], quantum approximate optimization algorithm (QAOA) [20], and more in [4]. Specifically, VQA uses *parametrized quantum models* to characterize loss functions, in particular those from quantum physics, which are potentially intractable for classical computing [26]. These parameters will then be optimized, usually through gradient-based approaches, to minimize the given loss function. Conventionally, parameterized quantum models are typically quantum circuits where each gate is parameterized by classical variables. Moreover, auto-differentiation techniques have been recently developed (e.g., [47, 59, 67]) for a scalable quantum gradient calculation on parameterized quantum circuits. We henceforth refer this existing framework as *differentiable digital quantum computing*.

Although parameterized quantum circuits are designed for NISQ applications, implementing (digital) quantum circuits still incurs non-negligible overheads, which significantly restricts the size of

---

[1]Pulse-level control is available on IBMQ computers.

36th Conference on Neural Information Processing Systems (NeurIPS 2022).

parametrized circuits that can be executed faithfully on NISQ machines. Moreover, the current parameterization in VQAs is also largely restricted by available parameterized gates and how they compose circuits, which in turn limits the expressive power of VQAs. A natural alternative to the current digital parameterization is to perform VQA directly on *parametrized analog signals (pulses)*, either on digital quantum machines with pulse-level controls, *or* on general analog quantum hardware. Indeed, parameterized analog pulses have the potential for more efficient NISQ implementation and better expressiveness as suggested in a recent perspective paper [43], which could be a more favorable parameterization for NISQ applications even when digital quantum gates are available.

However, there is not yet a systematic study of the analog parameterization for VQAs, its scalable training, and its quantitative benefits in achieving high-quality solutions in VQA applications.

**Contributions.** We conduct the first systematic study of the parameterization and scalable training of analog VQAs, which we call *differentiable analog quantum computing*. We also leverage our scalable training to demonstrate the quantitative benefits of the analog parameterization for specific VQA applications.

|  | **Diff. Quantum** | Neural ODE | Diff. Physics |
|---|---|---|---|
| $\frac{\mathrm{d}}{\mathrm{d}t}\mathbf{x}(t) =?$ | $-iH(\mathbf{v},t)\mathbf{x}(t)$ | $f(\mathbf{x}(t),\mathbf{v})$ | $f(\mathbf{x}(t),\mathbf{v})$ |
| Parametrization | Basis Function | Neural Network | Dynamics Equation |
| Forward | Time Evolution | Forward Pass | Time Integration |
| Backward | MC Integration | Autodiff | Auto/Manual Diff. |
| Device | Quantum | Classical | Classical |

Table 1: Machine learning in different dynamical systems. A diagram in Appendix A illustrates their connections.

Specifically, we formulate the general differentiable analog quantum computing as a control problem, $\min_{\mathbf{v}} \mathcal{L}(\mathbf{v})$, where the loss function $\mathcal{L}(\mathbf{v})$ is calculated from the $\mathbf{v}$-parametrized quantum state $\mathbf{x}(T; \mathbf{v})$ generated by quantum machines at the evolution time $t = T$. Intuitively, this control problem is no different from any classical ones except that the evolution of the quantum state in $[0, T]$ is governed by the Schrödinger equation

$$\frac{\mathrm{d}}{\mathrm{d}t}\mathbf{x}(t) = -iH(\mathbf{v}, t)\mathbf{x}(t), \tag{1}$$

where the Hamiltonian operator $H(\mathbf{v}, t)$ can be much more flexibly parameterized in $\mathbf{v}$ compared with parameterized quantum circuits (detailed in Section 3.2), and $i$ is the imaginary unit.

We also develop a scalable Monte Carlo integration technique of computing quantum-related gradients from the loss function $\mathcal{L}(\mathbf{v})$. A well-known difficulty in computing quantum-related gradients by classical means is the exponential cost associated with classical simulation of the quantum system. Thus, any scalable solution must leverage quantum machines to compute the gradients for themselves. Existing "auto-differentiation" techniques for parameterized quantum circuits (e.g., [47, 59, 67]) are designed for discrete-time evolution and the basic parameterized units (i.e., gates) are described by finite-dimensional matrices. Differential analog quantum computing exploits parameterization of continuous-time evolution and $H(\mathbf{v}, t)$ refers to a parameterized model from a functional space. Our Monte Carlo integration technique is designed to bridge this technical gap, which is later integrated with the stochastic gradient descent (SGD) for the entire framework for fast convergence and robustness against empirical noise. We further illustrate that our quantum stochastic gradient descent could still work with approximate descriptions of $H(\mathbf{v}, t)$ for domain applications.

An analogy could be drawn between our approach and classical deep learning, as summarized in Table 1. *Neural ODEs* [41, 10] view the structure of ResNet [27], $\mathbf{x}_{n+1} = \mathbf{x}_n + f(\mathbf{x}_n, \theta)$, as the solving of an ordinary differential equation,

$$\frac{\mathrm{d}}{\mathrm{d}t}\mathbf{x}(t) = f(\mathbf{x}(t), \mathbf{v}), \tag{2}$$

with $f(\cdot)$ as the network layer, $\mathbf{v}$ the network parameter, and $\mathbf{x}(t)$ the hidden state. This formula is similar to our system in (1), although we adopt a different parametrization than neural networks. Similarly, *differentiable physics* (e.g., [31]), which incorporates physical simulation into learning process, leverages existing numerical solvers and the autodiff functionality of deep learning with backpropagation to compute gradients of a physical or dynamical system, then integrates them into a neural network. This approach has proven to accelerate learning and generalization.

Differentiable analog quantum computing can be deemed as a special form of differentiable physics at the quantum scale. Similar to the promise of differentiable physics or neural ODEs, we have also observed the advantageous performance of differentiable analog quantum computing against the

conventional parameterized quantum circuits by orders of magnitude in major VQA applications like optimization (Section 4) and control (Section 5). Our auto-differentiation technique in quantum is however less efficient than classical ones that leverage the chain rule and the forward/backward propagation. This is due to the quantum no-cloning theorem [65] which prevents us from storing any arbitrary immediate state during quantum computation. So, we develop a sampling technique to compute gradients. To sum up, the key contributions of this work include:

- A formulation of differentiable, resource-efficient *analog* quantum computing framework (Sec. 3),
- A scalable technique of computing gradients by Monte Carlo sampling with SGD (Sec. 3.3),
- Formal analysis on correctness, efficiency, and robustness that showcases *exponential reduction of computational complexity* and bounded errors with approximate machine description (Sec. 3.4-3.6),
- Applications of our framework on quantum optimization (Sec. 4) and control (Sec. 5), with demonstrated advantages by *orders of magnitude* against parameterized quantum circuits. Our code is available here: `https://github.com/YilingQiao/diffquantum`

## 2   Related work

**Learning for dynamics and control**. Dynamical systems have widely been used to interpret and improve the design of machine learning algorithms. Compared to traditional discrete layers, Neural ODEs [24] demonstrate that continuous modeling of neural network can better learn the continuous structures [22, 57] with infinite depth [45] and constant memory cost [70]. Besides neural networks, differentiable physics [16] also computes analytic gradients of classical dynamical systems like rigid body [14, 53], articulated body [23, 64, 55], soft body [19, 21, 54], and fluids [62, 63, 29, 61]. Those methods have made great success on design [42], control [28] and system identification [38] tasks in the macroworld. Our framework, as the first differentiable dynamics for quantum computing, could have tremendous potential in chemistry and physics applications.

**Quantum machine learning & optimal control**. Quantum machine learning is a fast-developing emerging field (e.g., see the survey [5]) where variational quantum algorithms (VQAs) (e.g., see the survey [4] are one of the most promising candidates for NISQ applications. Quantum optimal control (succinctly, quantum control) aims to achieve a desired response from the quantum system by controlling the system parameters [18, 6]. Quantum control theory has empowered the growth of quantum technologies and found applications in several fields, ranging from molecular chemistry to quantum computing [56, 13]. The connection between quantum control theory and VQAs has been discussed recently [43, 46]. Several optimization algorithms have been developed to solve quantum control problems, including the Krotov method [49], monotonically convergence algorithms [69], non-iterative algorithms [68], the *GRadient Ascent Pulse Engineering (GRAPE)* algorithm [34], the *Chopped RAndom-Basis (CRAB)* algorithm [8], etc. The development of quantum computing also opens up the possibility of solving quantum control problems with quantum computers [17, 9]. These proposals take the approach of hybridizing quantum simulations with classical optimization routines. Nevertheless, they either do not support gradient-based methods or compute the gradients in a non-scalable way (e.g., via classical simulation), which significantly limits their performance especially comparing to ours. Existing pulse-level variational algorithms [39, 46, 40, 15] do not discuss their direct application to quantum analog machines and can not compute gradients analytically on quantum machines, while our paper resolves this problem.

## 3   Differentiable Analog Quantum Computing

Similar to classical dynamical systems, quantum systems also have states, governing equations, and observations, so there naturally exist plenty of optimization [48], control [12], and learning [5] problems for quantum computing. Inspired by the success of gradient-based approach in the classical domain, we propose a differentiable framework to compute the gradients of parametrized analog control signals on quantum computers, based on a "*forward simulation* with *stochastic sampling*" (see the workflow in Figure 1).

### 3.1   Quantum preliminaries

A *qubit* (or *quantum bit*) is the analogue of a classical bit in quantum computation. It is a two-level quantum-mechanical system described by vectors in the Hilbert space $\mathbb{C}^2$. We use Dirac notation

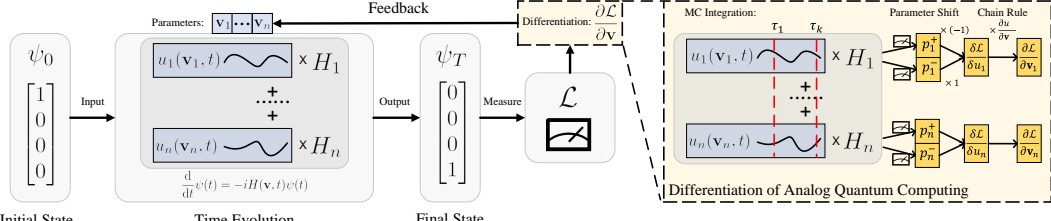

Figure 1: **The workflow of differentiable analog quantum computing.** In this 2-qubit example, the system starts with an initial ground state $\psi_0$ of dimension $4 = 2^2$ and evolves through the time following the Schrödinger equation (1). We control the dynamics of this quantum system by specifying the time-dependent Hamiltonian $H(\mathbf{v}, t)$, parameterized by trainable variables $\mathbf{v}$. In the end of the process, we measure the system and get a real-valued loss value, $\mathcal{L}$. The derivatives are computed as in the right box. Quantum computers cannot store computational graphs, so we propose to sample a time $t$ in the forward simulation and apply the parameter shift rule to compute the gradients. The derivatives are then used in the feedback loop to update $\mathbf{v}$ optimizing $\mathcal{L}$.

$|\psi\rangle$ to denote quantum states (i.e., unit vectors) $\psi$ in $\mathcal{H}$. For example, the classical "0" and "1" are represented by $|0\rangle = \begin{bmatrix} 1 \\ 0 \end{bmatrix}$ and $|1\rangle = \begin{bmatrix} 0 \\ 1 \end{bmatrix}$. One qubit states could be in any linear combination of $|0\rangle$ and $|1\rangle$, which is called *superposition*. An $n$-qubit state is a unit vector in the Kronecker tensor product $\otimes$ of $n$ single-qubit Hilbert spaces, i.e., $\mathcal{H} = \otimes_{i=1}^{n} \mathbb{C}^2 \cong \mathbb{C}^{2^n}$, whose dimension is exponential in $n$. For an $n$ by $m$ matrix $A$ and a $p$ by $q$ matrix $B$, their Kronecker product is an $np$ by $mq$ matrix where $(A \otimes B)_{pr+u, qs+v} = A_{r,s} B_{u,v}$. The **complex conjugate transpose** of $|\psi\rangle$ is denoted as $\langle\psi| = |\psi\rangle^\dagger$ († is the Hermitian conjugate). Therefore, the **inner product** of $\phi$ and $\psi$ could be written as $\langle\phi|\psi\rangle$.

The time evolution of quantum states under Schödinger equation is specified by the (time-dependent) Hermitian matrix $H(t)$ over the corresponding Hilbert space, known as the *Hamiltonian* of the quantum system. Typical single-qubit Hamiltonians include the famous *Pauli matrices*:

$$I = \begin{bmatrix} 1 & 0 \\ 0 & 1 \end{bmatrix}, \ X = \begin{bmatrix} 0 & 1 \\ 1 & 0 \end{bmatrix}, \ Y = \begin{bmatrix} 0 & -i \\ i & 0 \end{bmatrix}, \ Z = \begin{bmatrix} 1 & 0 \\ 0 & -1 \end{bmatrix}. \tag{3}$$

Similarly, a multi-qubit Hamiltonian can be built from the Pauli group consisting of tensor products of Pauli matrices. Conventionally we write $X_j$ ($Y_j$, $Z_j$) for a multi-qubit Hamiltonian to indicate the tensor product of multiple identity matrices while the $j$-th operand is $X$ ($Y$, $Z$), which represents operations on the $j$-th subsystem.

*Quantum measurement* refers to the procedure of extracting classical information from quantum systems. It is characterized by an Hermitian matrix $M$ called the *observable*. Measuring a quantum state $|\psi\rangle$ with observable $M$ is modelled as a random variable whose expectation value is $\langle\psi| M |\psi\rangle$.

## 3.2 Problem formulations

Most computational tasks in quantum simulation and control, like finding the ground state of a physics system, can be formulated as the following optimization problem. Given a quantum observable $M$ and an initial state $|\psi(0)\rangle = |\psi_0\rangle$, we seek for a parameter vector $\mathbf{v}$ by minimizing the loss function

$$\mathcal{L} = \langle\psi(T)| M |\psi(T)\rangle, \tag{4}$$

where the evolution of $|\psi(t)\rangle$ from $t = 0$ to $t = T$ subject to the Schrödinger equation (1). Here, $H(\mathbf{v}, t)$ is a Hamiltonian parametrized by $\mathbf{v}$ with form

$$H(\mathbf{v}, t) = H_c + \sum_{j=1}^{m} u_j(\mathbf{v}, t) H_j, \tag{5}$$

where $m$ is the number of control pulses, $H_c$ is a time-independent Hamiltonian (e.g. Pauli matrices), $H_j$ are tensor products of Pauli matrices, and the range of $u_j(\mathbf{v}, t)$ is $\mathbb{R}$. We also require $u_j(\mathbf{v}, t)$ to be differentiable with respect to $\mathbf{v}$ for any $t \in [0, T]$. The loss function $\mathcal{L}$ is hence *differentiable*. We can loosen the restriction of $H_j$ to general time-independent Hamiltonians if we have powerful enough quantum simulators. We keep it in this paper for the convenience of presenting our algorithm.

With specific $M$ and $|\psi_0\rangle$, optimization problem $\min_{\mathbf{v}} \mathcal{L}$ with post-processing can encode many essential classical and quantum problems. For example, any classical optimization of $f(x)$ over $n$-bit integers $x$ can be formulated as $M = \sum_x f(x) |x\rangle \langle x|$. More examples like Max-Cut problem and quantum state preparation are discussed in details in Sec. 4, 5.

**Abstract Quantum Analog Machines (AQAMs).** We propose AQAMs as our computational model. An AQAM optimizing the above loss function should be capable of consecutively: (1) evolving under $H(\mathbf{v}, t)$ for any $\mathbf{v}$ and time interval $[t_1, t_2] \subset [0, T]$; (2) evolving under constant Hamiltonian $H_j$ for time duration $\tau$ (effectively applying unitary transformation $e^{-iH_j\tau}$); (3) preparing $|\psi_0\rangle$; (4) measuring with observable $M$. For realistic quantum devices, we need to design AQAMs accordingly.

**Remark 3.1.** A trivial AQAM directly employs the Hamiltonian of a quantum device as $H(\mathbf{v}, t)$, parameterized by tunable pulses. In most realistic quantum devices, multi-qubit interactions are not tunable and weak compared to tunable single-qubit Hamiltonians. Thus $H_j$ can be simulated with high fidelity. Our method is robust against imprecise simulations of $H(\mathbf{v}, t)$ and $H_j$ (discussed in Section 3.6). Hence the trivial AQAMs are usually suitable for realistic quantum devices.

Our formulation of the problem via analog quantum computing is a generalization of the formulation via parameterized circuits. For example, a series of parameterized Pauli rotation gates $R_{P_j}(\theta_j) = \exp\{-i(\theta_j/2)P_j\}$ can be realized by $H(\mathbf{v}, t) = \sum_j \mathbf{v}_j \mathbf{1}_j(t)P_j$ with valuation $\mathbf{v}_j = \theta_j/2$, where $\mathbf{1}_j$ is the indicator function of $[j, j+1]$. However, simulating analog quantum computing via quantum circuits requires much longer time on nowadays devices [43], hence is unrealistic. So direct analog quantum computing can *exploit near-term quantum devices much better* than quantum circuits.

## 3.3 Quantum stochastic gradient descent

Our quantum SGD scheme for computing gradients on AQAMs is illustrated in this section, with its correctness, efficiency, and robustness discussed in the following sections.

We borrow the idea of mini-batches from classical SGD to deal with the gradients, and set two layers of mini-batches: (1) an integration mini-batch with size $b_{\text{int}}$; (2) an observation mini-batch with size $b_{\text{obs}}$. The integration mini-batch updates parameters according to the estimation of derivatives on the sampled time. The observation mini-batch repeats experiments to generate more precise measurement results. The scheme is displayed in Algorithm 1.

The forward and backward propagation of our SGD scheme is depicted in Figure 1. Notice that the inner loop is the only procedure on quantum machine. The difference of this procedure compared with the estimation of loss function $\mathcal{L}$ is the inserted evolution under $H_j$, which is beneficial in the error analysis in Section 3.6.

With the estimation of gradients, various optimizers designed for classical stochastic gradient descent are suitable to optimize the objective function. For example, Adam [35] is used in our experiments.

---

**Algorithm 1** Estimating gradients on an AQAM

---

**Classical inputs:** $b_{\text{int}}, b_{\text{obs}}$ (batch sizes), $m$ (number of control pulses), $u_j$ (parametrized pulses), $T$ (evolution duration), $\mathbf{v}$ (parameters)
**Quantum inputs:** $\mathcal{E}_{|\psi_0\rangle}$ (preparation of initial state), $H$ (parametrized system Hamiltonian), $H_j$ (pulse-controllable Hamiltonian), $\mathcal{E}_M$ (measurement procedure with observable $M$)
**Output:** a gradient estimation of $\mathcal{L}$ to $\mathbf{v}$

---

**for** $k \in \{1, ..., b_{\text{int}}\}$ **do**
    Draw $\tau \sim \text{Uniform}(0, T)$
    **for** $j \in \{1, ..., m\}, s \in \{-1, +1\}$ **do**
        **for** $l \in \{1, ..., b_{\text{obs}}\}$ **do**
            Prepare $|\psi_0\rangle$ via $\mathcal{E}_{|\psi_0\rangle}$
            Evolve under $H(\mathbf{v}, t)$ for time $[0, \tau]$
            Evolve under $H_j$ for duration $(1 + \frac{3}{4}s)\pi$
            Evolve under $H(\mathbf{v}, t)$ for time $[\tau, T]$
            $x_l \leftarrow$ observation sample from $\mathcal{E}_M$
        **end for**
        $\tilde{p}_j^{\text{sign}(s)} \leftarrow \frac{1}{b_{\text{obs}}} \sum_{l=1}^{b_{\text{obs}}} x_l$
    **end for**
    $\tilde{f}_k \leftarrow \sum_{j=1}^{m} \frac{\partial u_j}{\partial \mathbf{v}}(\mathbf{v}, \tau)(\tilde{p}_j^- - \tilde{p}_j^+)$
**end for**
$\widetilde{\frac{\partial \mathcal{L}}{\partial \mathbf{v}}} \leftarrow \frac{T}{b_{\text{int}}} \sum_{k=1}^{b_{\text{int}}} \tilde{f}_k$

---

## 3.4 Correctness of gradient estimation

We show that Algorithm 1 generates an unbiased estimation of the gradient $\frac{\partial \mathcal{L}}{\partial \mathbf{v}}$, and hence establishes its correctness. The proof of the following theorem is postponed to Appendix B.1.

**Theorem 3.2.** *The derivative of $\mathcal{L}$ to parameters $\mathbf{v}$ is*

$$\frac{\partial \mathcal{L}}{\partial \mathbf{v}} = \int_0^T \mathrm{d}\tau \; \sum_{j=1}^m \frac{\partial u_j}{\partial \mathbf{v}}(\mathbf{v}, \tau) \left(p_j^-(\tau) - p_j^+(\tau)\right). \tag{6}$$

*Here, $p_j^\pm(t) = \langle \psi_0 | U_\mathbf{v}^\dagger(t,0) e^{iH_j(1\pm\frac{3}{4})\pi} U_\mathbf{v}^\dagger(T,t) M U_\mathbf{v}(T,t) e^{-iH_j(1\pm\frac{3}{4})\pi} U_\mathbf{v}(t,0) | \psi_0 \rangle$, where $U_\mathbf{v}(t_2, t_1)$ denotes the time evolution operator for time interval $[t_1, t_2]$ under Hamiltonian $H(\mathbf{v}, t)$.*

One can interpret the above formula as a direct application of the chain rule over functional derivatives $\frac{\delta \mathcal{L}}{\delta u_j}$ and partial derivatives $\frac{\partial u_j}{\partial \mathbf{v}}$, since $\frac{\delta \mathcal{L}}{\delta u_j}(\mathbf{v}, t) = p_j^-(t) - p_j^+(t)$ by the parameter shift rule [47, 59], which is a technique for evaluating commutators of Hermitian by quantum processes.

Algorithm 1 estimates the integral in (6) via Monte Carlo integration (MCI) technique. We prove that the sampling of MCI has finite variances in Appendix B.1 when $u_j(\mathbf{v}, t)$ has bounded derivatives to $\mathbf{v}$. Hence MCI converges of rate $O(b_{\mathrm{int}}^{-\frac{1}{2}})$ [52]. Other numerical integral methods are also applicable here for different conditions on $u_j$ and $H_j$, and may have better convergence rates than MCI. We present it here because MCI has good generalization and simplicity. We also remark that similar ideas developed in this section have also appeared in [3] in the circuit model, whereas we develop everything in the analog quantum computing model.

Overall, the forward and backward propagation of differentiation of $\mathcal{L}$ are depicted in Figure 1. Since the parameterization of $u_j$ is arbitrary, a typical treatment is using a Fourier basis or a Legendre basis as the support of the parameterization. Neural networks are also suitable for pulse generation, whose gradients are easy to compute via backpropagation.

### 3.5 Scalability analysis

The resource consumption of our classical-quantum hybrid approach has two aspects: the classical and the quantum sides. The classical computation, as described in Algorithm 1, has $O(b_{\mathrm{int}} b_{\mathrm{obs}} m)$ numerical calculations. On the quantum side, the sampling process assessing the loss function and its derivatives takes $O(T)$ time each. The total running time on a quantum computer then is $O(b_{\mathrm{int}} b_{\mathrm{obs}} m T)$. Almost on every architecture of quantum devices, the number of control signals $m$ is *at most quadratic in the number of qubits $n$* (see some survey papers [36, 32, 7, 58]), showing excellent scalability of our approach. We exhibit the scalability of our method for up to 11 qubits in numerical experiments in Section 4.2. Our approach could in principle be readily applied to the most advanced existing quantum systems (e.g., [1] with $n \sim 60$). This is in sharp contrast to the case of GRAPE and CRAB algorithms, which rely on classical simulation of quantum systems with an exponential cost in $n$. For instance, the associated classical computation cost for $n \sim 60$ (i.e. at least $2^{60}$) is prohibitively high, almost reaching the limit of supercomputers today. *Our approach makes it feasible with only $O(n^2)$ complexity.*

### 3.6 Robustness analysis

In this section, we analyze systematic errors of gradient estimation produced of Algorithm 1. Our goal is to optimize the loss function assessed on a realistic and noisy quantum machine, whose capabilities of evolving under $H(\mathbf{v}, t)$ and $H_j$ are imperfect.

As a concrete example, when using the trivial AQAMs in Remark 3.1, the actual Hamiltonian of the quantum device may deviate from the description $H(\mathbf{v}, t)$ in our understanding, and the simulation of $H_j$ may be imprecise due to weak non-tunable terms in $H_c$. We show that our algorithm well approximates the gradient of loss function of the actual devices.

One major advantage of Algorithm 1 is that even though we have imperfect realization of Hamiltonian $H(\mathbf{v}, t)$ built in the AQAM, the quantum part is executed on the actual quantum machine following the accurate Hamiltonian $\widehat{H}(\mathbf{v}, t)$. As a result, the output of our algorithm well approximates the gradient for the actual quantum machine.

**Lemma 3.3.** *Let $\widehat{\frac{\partial \mathcal{L}}{\partial \mathbf{v}}}$ denote the accurate gradient of the loss function of the quantum machine, $\frac{\partial \mathcal{L}}{\partial \mathbf{v}}$ denote the estimated gradient via Algorithm 1, and $\|\cdot\|$ represents the matrix spectral norm [30], then*

$$\left| \frac{\partial \mathcal{L}}{\partial \mathbf{v}} - \widehat{\frac{\partial \mathcal{L}}{\partial \mathbf{v}}} \right| \leq 2\|M\| T \max_{\tau \in [0,T]} \left\| \frac{\partial H}{\partial \mathbf{v}}(\mathbf{v}, \tau) - \frac{\partial \widehat{H}}{\partial \mathbf{v}}(\mathbf{v}, \tau) \right\|.$$

The proof of this lemma is detailed in Appendix B.2. In other words, one can use Algorithm 1 to *optimize the control pulses without a precise understanding of the machine Hamiltonian*, if the difference $H(\mathbf{v}, t) - \widehat{H}(\mathbf{v}, t)$ varies slowly w.r.t. $\mathbf{v}$ (a rather mild condition to satisfy).

On the contrary, if one relies solely on the Hamiltonian $H(\mathbf{v}, t)$ built in the abstract quantum analog machine (e.g., the classical simulation of quantum systems in GRAPE or CRAB), the approximation error could be as large as the difference $H(\mathbf{v}, t) - \widehat{H}(\mathbf{v}, t)$ itself, a potentially large term compared with its derivative w.r.t. $\mathbf{v}$. An example is illustrated in Appendix B.3. This particular advantage of Algorithm 1 exists exactly because of its execution on the real machine, which *is potentially applicable in general scenarios where only approximate machine descriptions are obtainable*.

Similar to the circuit case, the systematic error caused by the imprecise evolution under $H_j$ is bounded by the error sum in each evolution under $H_j$ for duration $(1 \pm \frac{3}{4})\pi$, which is usually small.

# 4 Quantum optimization

Many important optimization problems in both physics and combinatorics that allow variational solutions can be formulated easily in our framework. For example, finding the ground state of physics systems can be solved by variational quantum eigensolver (VQE) (e.g. [50, 33]), and searching for the max-cut of graphs can be approximated by quantum approximate optimization algorithms (QAOA) [20]. Replacing parameterized circuits by AQAMs in existing variational quantum algorithms leads to significantly improved convergence based on numerical experiments on a classical simulator.

## 4.1 Variational quantum eigensolver

We exhibit our approach via an AQAM comparable to existing circuit VQE in terms of the pulse duration, with significantly better convergence and hardware efficiency in identifying the ground state of the $H_2$ molecule.

**Problem setting.** The Hamiltonian of $H_2$ molecule is expanded with Pauli matrices in the form $H_{H_2} = \alpha_0 \mathbb{I} + \alpha_1 Z_1 Z_2 + \alpha_2 X_1 X_2 + \alpha_3 Z_1 + \alpha_4 Z_2$, where $\alpha_i$ is a scalar weight calculated in [33]. The ground state $|\psi\rangle$ has the minimal energy, defined by $\mathrm{argmin}_{|\psi\rangle} \langle\psi| H_{H_2} |\psi\rangle$. This energy function is computed on real machines by Pauli measurements and linearity.

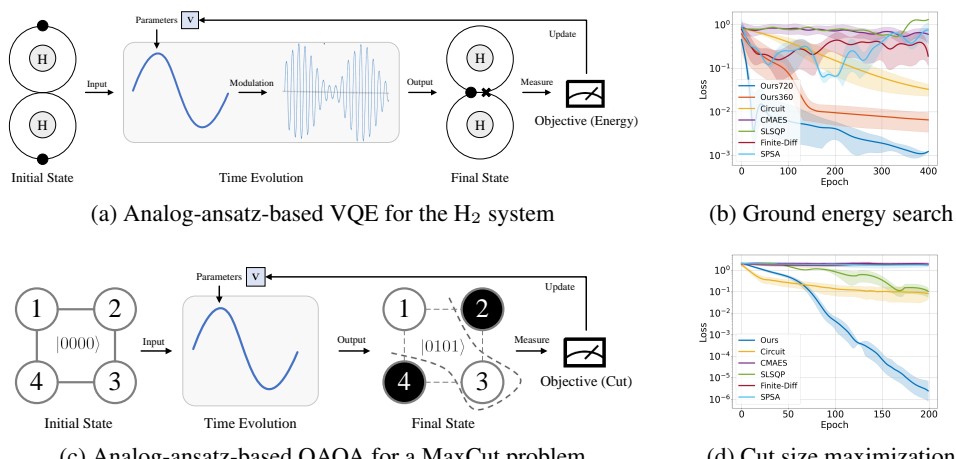

(a) Analog-ansatz-based VQE for the $H_2$ system

(b) Ground energy search

(c) Analog-ansatz-based QAOA for a MaxCut problem

(d) Cut size maximization

Figure 2: **Experiments on quantum optimization problems.** (a) Experiment results on the $H_2$ ground energy search. Loss function is the difference of the evaluated energy to the ground energy of $H_2$, and the lower is the better. Our method with the same (720dt) or less (360dt) pulse duration converges more than 10 times faster than the circuit model [33]. (b) Experiment results on finding the max cut for 4 vertices circular graph. Loss function is the difference of the evaluated cut size to the maximal cut size, the lower the better. Our method outperforms the others by orders of magnitude.

**AQAM design.** We propose an IBM-AQAM mimicking IBM transmon system [44] (a detailed introduction is given in Appendix C), and hence realizable on IBM's machines. The IBM-AQAM contains 2 qubits and 4 input pulses $u_{jk}(t)$, and can evolve under

$$H(t) = H_{\text{sys}} + \sum_{j,k\in\{0,1\}} \mathcal{M}_{jk}(u_{jk}, t) X_j. \tag{7}$$

Here $H_{\text{sys}}$ is a constant Hamiltonian. The input pulses to IBM quantum devices are $u_{jk}(t)$ which are complex values with norm less than 1. These pulses are modulated by the built-in modulation procedure $\mathcal{M}_{jk}$ of the IBM's quantum devices when executed on the real machine. Since the tunable terms have Hamiltonian $X_j$, we require the IBM-AQAM to be able to evolve under $X_j$. This is realizable on IBM's machines because in (7), $H_{\text{sys}}$ is much weaker than the microwave input pulses for each qubit in form $X_j$, ensuring a high fidelity simulation of Hamiltonian $X_j$ on real machines. We also require the IBM-AQAM to support initializing in state $|00\rangle$ and measuring with $M = H_{\text{H}_2}$, and these procedures are easy to implement for IBM's machines. We adopt the parameterization

$$u_{jk}(\mathbf{v}, t) = \mathcal{N}\left(\sum_{l=0}^{d}(\mathbf{v}_{jkl0} + i\mathbf{v}_{jkl1}) \cdot P_l\left(\frac{2t}{T} - 1\right)\right) \tag{8}$$

to make the pulse norms less than 1, where $\mathcal{N}(0) = 0, \mathcal{N}(z) = \frac{S(|z|)}{|z|}z$ for $z \neq 0$ is a differentiable normalization function restricting the norm, $S(x) = \frac{1-e^{-x}}{1+e^{-x}}$ is the shifted sigmoid function, $T$ is the duration, and $P_l$ is the $l$-th Legendre polynomial.

In [33], a parameterized circuit is proposed as layered tunable single qubit rotations and fixed cross-resonance gates, which are compiled to pulses fitting in (7) and sent to the IBM quantum devices. Their one-layer experiments over two qubits have cross resonance gates compiled to pulses with duration around 720dt where dt=0.22ns. We match it in our experiments, setting $T = 720$dt. Additionally, we test our approach with only half the duration, $T = 360$dt.

**Comparisons.** We compare our approach to circuit VQE, finite difference method, simultaneous perturbation stochastic approximation (SPSA), and derivative-free methods (CMAES [25] and SLSQP [37]) with the IBM-AQAM on a classical simulator. The experiment results are displayed in Figure 2b with a detailed hyper-parameter settings in Appendix D.1.1.

The circuit VQE converges to $\mathcal{L} = 0.02$, while our approach decreases lower: at 400 epoch, with 720dt it decreases to less than 0.002, and with 360dt it decreses to 0.01. In general, our approach is over 10 times more accurate than the circuit VQE, and 100 times more accurate than the derivative-free methods. The finite difference method and SPSA do not converge because of the intrinsic randomness of quantum measurements at relatively small observation mini-batch ($b_{\text{obs}} = 100$), which would be amplified by the small difference length. With a large enough observation mini-batch, the gradients evaluated by finite difference method has $\sim 3\%$ relative difference to the gradients evaluated by our approach. These results exhibit the advantages of our differentiable analog framework compared to circuit model and derivative-free analog models.[2]

### 4.2 Quantum approximate optimization algorithm

We also investigate the application of QAOA to approximate solutions for the MaxCut problem, an NP-complete problem. With a corresponding AQAM, we achieve a significantly better convergence.

**Problem setting.** Given a graph $G = \{E, V\}$ where $V$ is the vertex set and $E = \{(i, j)\}$ contains all the edges, our goal is to partition the vertices into two sets $(V_0, V_1)$ so that the number of edges between the two sets is maximized. A cut of an $n$-node graph $G$ is represented by an $n$-bit string $s = b_1 b_2 ... b_n$, with $b_j \in \{0, 1\}$ representing in which set the $j$-th vertex is. We use the computational basis $|s\rangle$ in an $n$-qubit register to represent the cut $s$. A maximum cut $|s\rangle$ maximizes the expected cut size $\langle s| C |s\rangle$, where $C = \frac{1}{2}\sum_{(j,k)\in E} C_{j,k}, C_{j,k} = \mathbb{I} - Z_j Z_k$. We test the performance on the circular graph shown in the leftmost of Figure 2c.

**AQAM design.** Farhi et al. [20] optimizes a $p$-layer circuit ansatz $U(\beta, \gamma) = \Pi_{j=1}^{p} e^{-i\beta_j B} e^{-i\gamma_j C}$, where $B = \sum_{j=1}^{n} X_j$. We set $p = 2$ as a baseline.

---

[2]We note that the benefits of using analog controls in VQE for $H_2$ molecule are also discovered in a recent result [46], where the optimization is conducted based on GRAPE-like techniques.

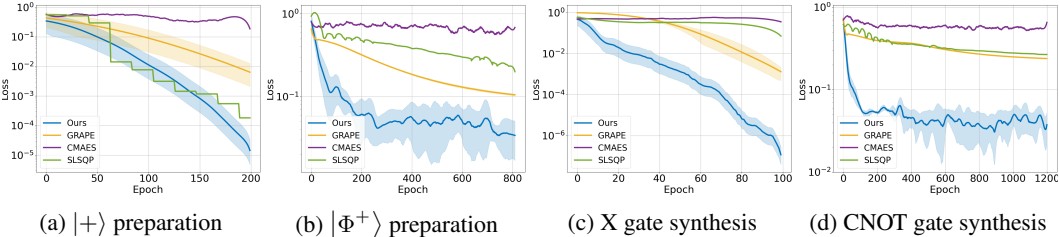

| (a) $|+\rangle$ preparation | (b) $|\Phi^+\rangle$ preparation | (c) X gate synthesis | (d) CNOT gate synthesis |

Figure 3: **Quantum Control.** We apply our differentiable analog quantum computing framework to state preparation and gate synthesis. Four methods (SLSQP, CMAES, GRAPE, and ours) are used in these tasks. In gate synthesis, the accuracy of the built-in gates from IBM Qiskit is shown in dashed lines. Our method outperforms the other three algorithms in terms of convergence rate and accuracy by up to orders of magnitude.

To have a fair comparison, we design a Cut-AQAM corresponding to the above circuit: $H(t) = \frac{1}{2\pi} \sum_{j=1}^{4} \left( u_{j0}(t)C_{j,j+1} + u_{j1}(t)X_j \right)$.

The input pulses are real functions $u_{jk}(t)$, where we restrict the energy input by requiring $|u_{jk}| \leq 1$. Cut-AQAM natively supports evolving merely under $C_{j,j+1}$ or $X_j$ by setting $u_{jk}$ as indicator functions. We also require it to support initializing in state $|0\rangle$ and measuring with observable $C$. In our experiment, we set the duration $T = 4$ within which the circuit QAOA can be realized by Cut-AQAM. Similar to (8), we parameterize $u_{jk}(\mathbf{v}, t)$ by a normalized linear combination of Legendre's polynomial.

**Comparisons.** The experiments are set up in a similar way as in Section 4.1, with details in Appendix D.1.2. Results are shown in Figure 2d. The circuit QAOA and SLSQP converges to 0.08 at 200 epoch, while our method converges to $2.6 \times 10^{-6}$. Finite difference method, SPSA, and CMAES do not converge to a value less than 1. The analysis is similar to Section 4.1. Since Cut-AQAM does not have a high frequency modulation, SLSQP also converges close to 0, but is slower than our approach. We conduct larger experiments for up to 11 qubits, which shows good scalability and better performance compared to circuit QAOA. For details, see Figure 5 in Appendix D.1.2.

## 5 Quantum Control

Quantum control problems fall into two categories: 1) *state preparation*: to steer a given initial state into a target final state; 2) *gate synthesis*: to effect a specific unitary transformation (quantum gate) in the system. In what follows, we discuss how to formulate and solve quantum control problems using our differentiable analog quantum computing framework. We demonstrate our methodology by numerical experiments on the IBM-AQAM.

### 5.1 State Preparation

To prepare the target state $|\psi_{tar}\rangle$ from certain fixed initial state, we desire a parameter vector $\mathbf{v}$ that minimizes the loss function defined in (4) with the observable $M = \mathbb{I} - |\psi_{tar}\rangle \langle\psi_{tar}|$. We consider two tasks: 1) to prepare the state $|+\rangle$ from $|0\rangle$; 2) to prepare the two-qubit maximally entangled state $|\Phi^+\rangle$ from $|00\rangle$.[3] In both tasks, the loss function is readily computed as the measurement merely involves local Pauli operators (see Appendix D.2.1), and can be carried out on the IBM-AQAM.

In the numerical experiments, the pulse duration is fixed as $T = 20\text{dt}$ for the $|+\rangle$ state, and $T = 1200\text{dt}$ for the $|\Phi^+\rangle$ state. We also compare our method with two gradient-free methods (SLSQP, CMAES) and the GRAPE algorithm. In both tasks, ours achieves faster convergence than all other three methods. In the $|+\rangle$ task, the final loss value from our method reads approximately $10^{-5}$, which is 18 times better than the second best result (i.e., SLSQP), as in in Figure 3 (a). In the $|\Phi^+\rangle$ task, the final loss value from our method reads 0.034, which is 3 times better than GRAPE and 6 times better than SLSQP, as shown in Figure 3 (b).

---

[3] $|+\rangle = \frac{1}{\sqrt{2}}(|0\rangle + |1\rangle)$ and $|\Phi^+\rangle = \frac{1}{\sqrt{2}}(|00\rangle + |11\rangle)$.

## 5.2 Gate Synthesis

Gate synthesis is more challenging than state preparation because we hope to engineer a full unitary gate $U_{tar}$ instead of just mapping a single state to a target state. To this end we first specify a set of pairs $\mathcal{S} = \{(|\mathbf{x}_j\rangle, |\mathbf{y}_j\rangle)\}_{j=1}^k$ that completely determines $U_{tar}$ in the sense that no unitary map $U$ other than $U_{tar}$ satisfies $\left|\langle \mathbf{y}_j| U |\mathbf{x}_j\rangle\right| = 1$ for all $j = 1, 2, .., k$. Then, we consider the loss function $\mathcal{L} = \frac{1}{k}\sum_{j=1}^k \mathcal{L}_j$, where each $\mathcal{L}_j$ is defined as in (4) with quantum observable $M = \mathbb{I} - |\mathbf{y}_j\rangle\langle \mathbf{y}_j|$ and initial state $|\mathbf{x}_j\rangle$. When $\mathcal{L}(\mathbf{v})$ is close to 0, the time evolution controlled by the parameter $\mathbf{v}$ is approximately $U_{tar}$.

The X gate and CNOT gate are widely used in digital quantum computing and supported by IBM Qiskit. We now exemplify our method by recovering these two gates on the IBM-AQAM. In both cases, the loss function can be readily computed on IBM machines. See Appendix D.2.2 for detail.

The pulse duration is chosen to be comparable to the built-in ones in IBM Qiskit: $T = 160\texttt{dt}$ for X gate, and $T = 1200\texttt{dt}$ for CNOT gate.[4] We apply four methods (SLSQP, CMAES, GRAPE, ours) to the gate synthesis tasks. The results are shown in Figure 3 (c), (d). In the synthesis of X gate, the final loss value from our method is $1.17 \times 10^{-7}$, which is over $10^4$ times more accurate than the other three methods. In the more involved task of synthesizing CNOT gate, our method returns a pulse sequence with loss $0.0172$, while the results from other methods are no less than $0.2$. It is worth noting that the loss of IBM built-in calibrated pulses evaluated on IBM machines are $0.019$ for X gate and $0.043$ for CNOT gate when no measurement error mitigation or state preparation error mitigation technique is applied.

# 6 Conclusion and Future Work

We have introduced the first differentiable analog quantum computing framework with a quantum stochastic gradient descent algorithm that allows directly optimizing the analog pulse control signal on quantum computers. Since the computation history in a quantum system cannot be stored or reused for the purpose of computing gradient, we construct a novel formulation for derivatives on quantum computers based on a forward pass with Monte Carlo sampling. With the proposed algorithm, our method outperforms prior methods by orders of magnitude with better hardware efficiency on quantum optimization and control tasks.

**Generalization.** This work suggests a larger scope of differentiable physics that extends beyond classical cases to quantum scale, where insights and techniques from one scale could potentially be translated to the other. For instance, our quantum differentiable framework is largely inspired by the classical counterpart. We also believe some of our findings for quantum computing could also be generally applicable in the classical paradigm: e.g., our differentiation technique is likely extendable to *general linear dynamical systems* and our robustness analysis also holds in general when *only approximate machine descriptions are known*.

**Limitations and Future Work.** As a first step, we only describe our framework with a few important, but small, demonstrating examples running on simulators. The great promise of our framework lies in implementing large-scale VQA applications on real quantum machines and bringing useful quantum applications to practice. To that end, we plan to incorporate real-world quantum machines into our framework, and solve large-scale VQA tasks.

# Acknowledgement

We thank the anonymous reviewers for their helpful feedback. This work was funded in part by U.S. Army Research Office DURIP grant, U.S. Department of Energy, Office of Science, Office of Advanced Scientific Computing Research Award Number DE-SC0019040 and DE-SC0020273, Air Force Office of Scientific Research under award number FA9550-21-1-0209, U.S. National Science Foundation grant CCF-1816695 & CCF-1942837 (CAREER), Barry Mersky and Capital One E-Nnovate Endowed Professorships.

---

[4]In IBM Qiskit, the X gate is implemented by a pulse of duration $T = 160\texttt{dt}$, and CNOT is by $T = 1056\texttt{dt}$.

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
