# Appendix

## A  Connection between Our Method and Deep Learning

We show the similarities between our method, Neural ODE, and differentiable physics in Figure 4. All the three approaches have a differentiable system governed by some kinds of differential equations. Our method parametrizes the dynamics using continuous basis functions; Neural ODE uses neural networks; and Differentiable physics describes the dynamics system using physics equations like Newton's Second Law, Navier–Stokes equations.

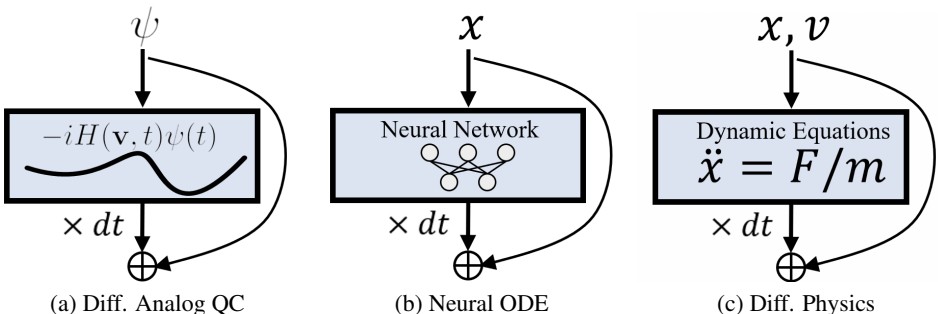

(a) Diff. Analog QC        (b) Neural ODE        (c) Diff. Physics

Figure 4: Connections between Diff. Analog computing and other deep learning models.

## B  Theory

### B.1  Derivation of gradient

**Lemma B.1.** *The derivative of the time evolution operator is*

$$\frac{\partial U_{\mathbf{v}}(t_2, t_1)}{\partial \mathbf{v}} = -i \int_{t_1}^{t_2} \mathrm{d}\tau \, U_{\mathbf{v}}(t_2, \tau) \frac{\partial H(\mathbf{v}, \tau)}{\partial \mathbf{v}} U_{\mathbf{v}}(\tau, t_1) \tag{9}$$

*Proof.* Let $U_{\mathbf{v}}(t_2, t_1)$ be as defined in Theorem 3.2. We can re-write (1) in the form of time evolution operator,

$$i \frac{\partial U_{\mathbf{v}}(t, 0)}{\partial t} = H(\mathbf{v}, t) U(t, 0). \tag{10}$$

It follows that

$$
\begin{aligned}
& i \left( U_{\mathbf{u}}(t, 0) - U_{\mathbf{v}}(t, 0) \right)' \\
={}& H(\mathbf{u}, t) U_{\mathbf{u}}(t, 0) - H(\mathbf{v}, t) U_{\mathbf{v}}(t, 0) \\
={}& [H(\mathbf{u}, t) - H(\mathbf{v}, t)] U_{\mathbf{u}}(t, 0) + H(\mathbf{v}, t)(U_{\mathbf{u}} - U_{\mathbf{v}})(t, 0).
\end{aligned} \tag{11}
$$

By the variation-of-parameters formula [11], we have

$$U_{\mathbf{u}}(t, 0) - U_{\mathbf{v}}(t, 0) = -i \int_0^t \mathrm{d}\tau \, U_{\mathbf{v}}(t, \tau)[H(\mathbf{u}, \tau) - H(\mathbf{v}, \tau)] U_{\mathbf{u}}(\tau, 0). \tag{12}$$

Now, if $\mathbf{u} = \mathbf{v} + h$, we have $H(\mathbf{u}, t) = H(\mathbf{v}, t) + h \frac{\partial H(\mathbf{v}, t)}{\partial \mathbf{v}} + O(h^2)$. Therefore, for each $t \geq 0$, we have

$$\frac{U_{\mathbf{u}}(t, 0) - U_{\mathbf{v}}(t, 0)}{h} = -i \int_0^t \mathrm{d}\tau \, U_{\mathbf{v}}(t, \tau) \frac{\partial H(\mathbf{v}, \tau)}{\partial \mathbf{v}} U_{\mathbf{u}}(\tau, 0) + O(h), \tag{13}$$

which implies the desired result (9) by taking $h \to 0$. $\qquad\square$

**Proposition B.2.** *Let $\mathcal{L}$ be defined as* (4), *and* $H(\mathbf{v}, t) = \sum_j f_j(\mathbf{v}, t) H_j$.

$$\frac{\partial \mathcal{L}}{\partial \mathbf{v}} = \left( -i \int_0^T d\tau \sum_{j=1}^m \frac{\partial f_j(\mathbf{v}, \tau)}{\partial \mathbf{v}} \langle \psi(\tau) | M_\tau H_j | \psi(\tau) \rangle \right) + h.c., \tag{14}$$

*where* $|\psi(\tau)\rangle = U(\tau, 0) |\psi(0)\rangle$ *is the state evolving under* $H(\mathbf{v}, t)$ *to time* $\tau$ *starting from* $|\psi(0)\rangle$, *and* $M_\tau = U^\dagger(T, \tau) M U(T, \tau)$. *Here h.c. means the Hermitian conjugate of the first half of the expression (this is a common abbreviation among physics, resulting in a Hermitian matrix).*

*Proof.* The derivatives can be computed as,

$$\frac{\partial \mathcal{L}}{\partial \mathbf{v}} = \langle \psi(0) | U^\dagger(T, 0) M \frac{\partial U(T, 0)}{\partial \mathbf{v}} | \psi(0) \rangle + h.c.$$

$$= \left( -i \int_0^T d\tau \sum_j \frac{\partial f_j(\mathbf{v}, \tau)}{\partial \mathbf{v}} \langle \psi(0) | U^\dagger(\tau, 0) U^\dagger(T, \tau) M U(T, \tau) H_j U(\tau, 0) |\psi(0)\rangle \right) + h.c.$$

$$\tag{15}$$

$\square$

To compute the derivative $\frac{\partial \mathcal{L}}{\partial \mathbf{v}}$ on a quantum machine, we invoke the parameter-shift formula.

Then we show the MCI generates an unbiased estimation of $\frac{\partial \mathcal{L}}{\partial \mathbf{v}}$ and converges of rate $O(b_{\text{int}}^{-\frac{1}{2}})$. We require that the derivatives of $u(\mathbf{v}, t)$ are bounded: $\forall t \in [0, T]$, $\left| \frac{\partial u_j(\mathbf{v}, t)}{\partial \mathbf{v}} \right| \leq D$.

The integration mini-batch draws time samplings $\tau \sim \text{Uniform}(0, T)$, and evaluates

$$f(\tau) = \sum_{j=1}^m \frac{\partial u_j}{\partial \mathbf{v}}(\mathbf{v}, \tau) \left( p_j^-(\tau) - p_j^+(\tau) \right). \tag{16}$$

Then (6) turns to $\frac{\partial \mathcal{L}}{\partial \mathbf{v}} = \int_0^T f(\tau) d\tau$. By MCI, the average value of $T \cdot f(\tau)$ in the integration mini-batch is an unbiased estimation of $\frac{\partial \mathcal{L}}{\partial \mathbf{v}}$. By applying the Popocivius's inequality and the Cauchy-Schwarz inequality, we obtain the following variance bound, which guarantees the low error of MCI.

**Proposition B.3.** *The variance of* $f(\tau)$ *is finite. Specifically,*

$$Var[f(\tau)] \leq 4m^2 \|M\|^2 D^2. \tag{17}$$

## B.2 Proof of Lemma 3.3

*Proof.* We let $\widehat{\frac{\partial \mathcal{L}}{\partial \mathbf{v}}}$ denote the accurate gradient of the loss function of the quantum machine, and $\frac{\partial \mathcal{L}}{\partial \mathbf{v}}$ denote the estimated gradient via Algorithm 1. Their analytical expressions are

$$\frac{\partial \mathcal{L}}{\partial \mathbf{v}} = -i \int_0^T d\tau \langle \psi_0 | \widehat{U}_\mathbf{v}^\dagger(T, 0) M \widehat{U}_\mathbf{v}(T, \tau) \frac{\partial H}{\partial \mathbf{v}} \widehat{U}(\tau, 0) |\psi_0\rangle + h.c., \tag{18}$$

$$\widehat{\frac{\partial \mathcal{L}}{\partial \mathbf{v}}} = -i \int_0^T d\tau \langle \psi_0 | \widehat{U}_\mathbf{v}^\dagger(T, 0) M \widehat{U}_\mathbf{v}(T, \tau) \frac{\partial \widehat{H}}{\partial \mathbf{v}} \widehat{U}(\tau, 0) |\psi_0\rangle + h.c., \tag{19}$$

where $h.c.$ means the Hermitian conjugate of the first half of the expression, and $\widehat{U}_\mathbf{v}(t_2, t_1)$ is the evolution operator under the actual Hamiltonian $\widehat{H}(\mathbf{v}, t)$ of the realistic machine. Hence the difference between these two evaluations are

$$\left| \frac{\partial \mathcal{L}}{\partial \mathbf{v}} - \widehat{\frac{\partial \mathcal{L}}{\partial \mathbf{v}}} \right| = \left| -i \int_0^T d\tau \langle \psi_0 | \widehat{U}_\mathbf{v}^\dagger(T, 0) M \widehat{U}_\mathbf{v}(T, \tau) \left( \frac{\partial H}{\partial \mathbf{v}} - \frac{\partial \widehat{H}}{\partial \mathbf{v}} \right) \widehat{U}(\tau, 0) |\psi_0\rangle + h.c. \right| \tag{20}$$

$$\leq 2T \|M\| \max_{\tau \in [0, T]} \left\| \frac{\partial H}{\partial \mathbf{v}}(\mathbf{v}, \tau) - \frac{\partial \widehat{H}}{\partial \mathbf{v}}(\mathbf{v}, \tau) \right\|, \tag{21}$$

where $\|\cdot\|$ is the spectral norm [30]. $\square$

### B.3 Simulating Approximated System

We show that there are cases that, when simulating the system based on imprecise approximation of the machine Hamiltonian, the gradient estimated is largely off to the actual gradient.

Consider a 1 qubit system $H(v,t) = \frac{\pi}{4}Y + vY$ evolve for time in $[0,1]$ with initial state $|\psi_0\rangle = |0\rangle$, and measurement $M = |0\rangle\langle 0|$. Let the approximation be $\tilde{H}(v,t) = vY$, whose difference to $H(v,t)$ is constant. Then by Algorithm 1 where the system evolution is executed on quantum machine, the estimated gradient is exact since $\frac{\partial H}{\partial \mathbf{v}} = \frac{\partial \tilde{H}}{\partial \mathbf{v}} = Y$. When simulating the system with $\tilde{H}$ and estimating the gradient as $\widehat{\frac{\partial \mathcal{L}}{\partial \mathbf{v}}}$, however, consider when $v = 0$. Note

$$\widehat{\frac{\partial \mathcal{L}}{\partial \mathbf{v}}} = -i\langle\psi_0| MY |\psi_0\rangle + h.c. = 0, \tag{22}$$

$$\frac{\partial \mathcal{L}}{\partial \mathbf{v}} = -i\langle\psi_0| e^{i\frac{\pi}{4}Y} MY e^{-i\frac{\pi}{4}Y} |\psi_0\rangle + h.c. = 1. \tag{23}$$

In a similar way, one can construct cases where the difference between estimated gradient based on the approximated system and actual gradient is as large as $\Theta(T \cdot \|M\| \cdot \|H(v,t) - \tilde{H}(v,t)\|)$.

## C  IBM Pulse-level control

We adopt the following effective Hamiltonian [44] to model the pulse-level qubit control on the $n$-qubit IBM machine:

$$H(t) = H_{sys} + H_{ctrl}(t). \tag{24}$$

The system (drift) Hamiltonian is independent of time,

$$H_{sys} = \sum_{j=1}^{n} \frac{\epsilon_j}{2}(I_j - Z_j) + \sum_{\substack{(j,k)\in E \\ j\neq k}} \frac{J_{jk}}{4}(X_j X_k + Y_j Y_k), \tag{25}$$

where $E$ is a bidirectional connectivity graph with self loops, and $J_{jk} = J_{kj}$. The control Hamiltonian is

$$H_{ctrl}(t) = \sum_{j=1}^{n} \sum_{k\in E_j} \mathcal{M}_{jk}(u_{jk}, t) X_j., \tag{26}$$

where the function $\mathcal{M}_{jk}(u_{jk}, t)$ modulates pulse $u_{jk}$ with a local oscillatory frequency $\omega_k$ of qubit $k$:

$$\mathcal{M}_{jk}(u_{jk}, t) = \Omega_j \mathbf{Re}\{e^{i\omega_k t} u_{jk}(t)\}, \tag{27}$$

where $\Omega_j$ is the maximal energy input on qubit $j$, $u_{jk}(t)$ is a complex-valued control pulses and $|u_{jk}(t)| \leq 1$.

On the real machine, $u_{jk}(t)$ should be a piece-wise constant complex-valued function, with each piece having length `dt` $= 0.222$ns. In our simulation we take $u_{jk}(t)$ as continuous function. Taking piece-wise constant approximation of our results will generate pulses that are available on the real machine.

In our 1-qubit experiments, the involved constants are: $n = 1$, $\epsilon_1 = 3.29 \times 10^{10}$, $\omega_1 = 2\pi \times 5.23 \times 10^9$, $\Omega_1 = 9.55 \times 10^8$ and $E_1 = \{1\}$.

In our 2-qubit experiments, the involved constants are: $n = 2$, $\epsilon_1 = 3.29 \times 10^{10}$, $\epsilon_2 = 3.15 \times 10^{10}$, $\omega_1 = 2\pi \times 5.23 \times 10^9$, $\omega_2 = 2\pi \times 5.01 \times 10^9$, $\Omega_1 = 9.55 \times 10^8$, $\Omega_2 = 9.87 \times 10^8$, $J_{12} = 1.23 \times 10^7$ and $E_1 = E_2 = \{1,2\}$.

## D  Experiment Details

We present the detailed experiment setups here. Our code is written in Python and C++ and the experiments run on a Desktop with an Intel Xeon W-2123 CPU @ 3.6GHz. In the comparisons, we use existing packages for CMAES and SLSQP with default hyperparameters. More details can be found in our supplementary code.

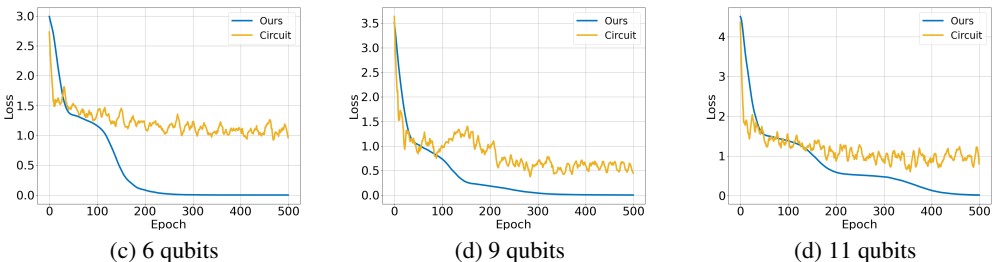

| (c) 6 qubits | (d) 9 qubits | (d) 11 qubits |

Figure 5: **Comparison with circuit QAOA in scaling up MaxCut problems.** As the size of graphs increases, the number of qubits increases and ours still converges better than VQAs.

## D.1 Quantum Optimization

### D.1.1 Experiment details in comparison to VQE

The workflow of analog-ansatz-based VQE is shown in Figure 2a. The $H_2$ molecule Hamiltonian at bond distance has coefficients $\alpha_0 = -1.0524, \alpha_1 = -0.0113, \alpha_2 = 0.1809, \alpha_3 = -0.3979, \alpha_4 = 0.3979$.

The parameters of different methods are presented below. When updating, the measurements for all methods has sampling size (observation mini-batch size) $b_{\text{obs}} = 100$. When evaluating, matrix multiplication is utilized to have accurate loss function calculation.

**Hyper-parameters in our approach.** The integration mini-batch size $b_{\text{int}} = 6$; the maximal degree of Legendre polynomial is $d = 3$; the initialization uses Gaussian distribution centered at 0; the optimizer is Adam with learning rate $lr = 0.02$; time duration is $T = 720\text{dt}$ and $T = 360\text{dt}$ correspondingly.

**Circuit ansatz VQE:** We use a three layered circuit ansatz. Let $U_j(\theta_1, \theta_2, \theta_3) = e^{-i\frac{\theta_3}{2}Z_j}e^{-i\frac{\theta_2}{2}X_j}e^{-i\frac{\theta_1}{2}Z_j}$, and $V_j(\theta) = U_1(\theta_{j11}, \theta_{j21}, \theta_{j31})U_2(\theta_{j12}, \theta_{j22}, \theta_{j32})$. Then the circuit ansatz is $U(\theta) = V_2(\theta)e^{-i\frac{\pi}{2}Z_1X_2}V_1(\theta)$. We substitute a ZX rotation for the cross resonance gate in [33] since they generate the same entanglement.

**Finite difference:** We use the same ansatz as in our approach. The finite difference formula used is:

$$FD\left(\frac{\partial\mathcal{L}}{\partial\mathbf{v}_i}, h\right) = \frac{1}{2h}\left(\mathcal{L}(\mathbf{v} + h\mathbf{e}_i) - \mathcal{L}(\mathbf{v} - h\mathbf{e}_i)\right), \tag{28}$$

where $\mathbf{e}_i$ is the unit vector whose only non-zero entry is the $i$-th entry with value 1. We take a small pertubation $h = 10^{-4}$ and simulate the quantum system twice to evaluate corresponding influences.

### D.1.2 Experiment details in comparison to QAOA

The workflow of analog-ansatz-based QAOA is shown in Figure 2c. The settings of the measurements and the finite difference method are similar as Appendix D.1.1.

**Hyper-parameters in our approach:** The integration mini-batch size $b_{\text{int}} = 1$; the maximal degree of Legendre polynomial is $d = 3$; the initialization uses Gaussian distribution centered at 0; the optimizer is Adam with learning rate $lr = 0.02$; time duration is $T = 4$.

We also conduct larger experiments on cycle graph involving $6, 9, 11$ vertices, as in Figure 5. Our method still outperforms circuit QAOA with better convergence rate and lower final loss.

## D.2 Quantum Control

### D.2.1 Experiment details in state preparation

We implement our method, together with three other methods (SLSQP, CAMES, GRAPE) to the two state preparation tasks: (a) to prepare $|+\rangle$ from $|0\rangle$; (b) to prepare $|\Phi^+\rangle$ from $|00\rangle$.

**Quantum measurement.** In both tasks, the quantum observable used in computing the loss function can be expressed as a sum local Pauli operators. This nice property makes the loss function easy to evaluate on the quantum computer. For the state $|+\rangle$, the observable is

$$M = \mathbb{I} - |+\rangle\langle+| = \frac{1}{2}(\mathbb{I} - X). \tag{29}$$

Similarly, we compute the observable used in preparing the $|\Phi^+\rangle$ state:

$$M = \mathbb{I} - |\Phi^+\rangle\langle\Phi^+| = \frac{1}{4}(X_1 X_2 - Y_1 Y_2 + Z_1 Z_2 - 3\mathbb{I}). \tag{30}$$

**Hyper-parameters in our approach.** The integration mini-batch size is $b_{int} = 1$ in task (a), and $b_{int} = 400$ in task (b). The maximal degree of Legendre polynomial is $d = 4$ in both cases. The initialization uses Gaussian distribution centered at $0$. The optimizer is Adam in both tasks, with learning rate $lr = 0.01$ in (a) and $lr = 0.05$ in (b).

### D.2.2 Experiment details in gate synthesis

We implement our method, together with three other methods (SLSQP, CAMES, GRAPE) to the two gate synthesis tasks: (a) to synthesize X gate; (b) to synthesize CNOT gate. The matrix representation of the X gate can be found in (3). The CNOT gate is shown below:

$$\text{CNOT} = \begin{bmatrix} 1 & 0 & 0 & 0 \\ 0 & 1 & 0 & 0 \\ 0 & 0 & 0 & 1 \\ 0 & 0 & 1 & 0 \end{bmatrix}. \tag{31}$$

**Quantum measurement.** We identify two sets $\mathcal{S}_X$ and $\mathcal{S}_{CNOT}$ that completely determine the X gate and CNOT gate correspondingly:

$$\mathcal{S}_X = \{(|0\rangle, |1\rangle), (|1\rangle, |0\rangle), (|+\rangle, |+\rangle)\}, \tag{32}$$
$$\mathcal{S}_{CNOT} = \{(|00\rangle, |00\rangle), (|01\rangle, |01\rangle), (|10\rangle, |11\rangle), (|11\rangle, |10\rangle), (|++\rangle, |++\rangle)\}. \tag{33}$$

In $\mathcal{S}_X$ and $\mathcal{S}_{CNOT}$, one of the computational basis pair can be safely removed. But for the convenience of training we keep them in our experiments.

**Hyper-parameters in our approach.** The integration mini-batch size is $b_{int} = 1$ in task (a), and $b_{int} = 400$ in task (b). The maximal degree of Legendre polynomial is $d = 4$ in both cases. The initialization uses Gaussian distribution centered at $0$. The optimizer is Adam in both tasks, with learning rate $lr = 0.005$ in (a) and $lr = 0.03$ in (b).