# OpenReview forum: "Differentiable Analog Quantum Computing for Optimization and Control"
_NeurIPS.cc/2022/Conference — NeurIPS 2022 Accept_

### Official Review · Reviewer_cLaN · 2022-07-07

**Rating:** 7
**Confidence:** 1
**Soundness:** 4 excellent
**Presentation:** 4 excellent
**Contribution:** 4 excellent

**Summary:**

This papers introduces the first differentiable analog quantum computing framework with a quantum SGD algorithm which optimizes the analog pulse control singla on quantum computers.

**Questions:**

Sorry, I don't know enough in the field to ask relevant questions and initiate fruitful discussions.

**Limitations:**

No negative societal impact

**Strengths And Weaknesses:**

Strengths:
- The paper looks serious and does what it claims to achieve.
- The benchmarks are convincing.
- I liked the fact that there was a robustness analysis.
- The paper is well written and structured.

Weaknesses:
- I am not knowledgeable enough in the field to spot weaknesses.

---

> ### Author Response · Authors · 2022-08-02
> **Response**
>
> We sincerely appreciate your time reading and reviewing our paper.
>
> In this paper, we are 1) proposing to use continuous-time representations for quantum computing; 2) designing an auto differentiation mechanism on quantum computers that can compute gradients for our representations; 3) demonstrating the correctness, effectiveness, and robustness of our framework with both theoretical and empirical analysis.
>
> The continuous-time representations used in our paper are currently basis functions. But for example, it could possibly be neural networks in future works. We hope this paper could help machine learning and quantum computing (QC) to better benefit each other, where more ML techniques can be introduced to QC and QC can solve more ML problems with its computational resources. Thanks again for your kind support. Please feel free to let us know if you have any questions or concerns during the discussion period.

---

### Official Review · Reviewer_gTD7 · 2022-07-10

**Rating:** 5
**Confidence:** 5
**Soundness:** 3 good
**Presentation:** 3 good
**Contribution:** 2 fair

**Summary:**

The paper studies the problem of variational quantum algorithms and proposes to tune the parameters on the pulse level to achieve better efficiency than optimizations on the gate level. Results on both quantum optimization (QAOA) and quantum control show large advantages.


**Questions:**

1. What are the core differences to the previous pulse-based variational learning methods besides the parameter shift method to obtaining gradients?

2. Figure 1 caption 'an' initial

**Ethics Review Area:**

["I don’t know"]

**Limitations:**

No significant negative societal impact.

**Strengths And Weaknesses:**

Strengths:
1. Good motivation of optimizations the algorithms on the pulse level to better leverage the current quantum machines.
2. Good explanations on the parameter shift method to obtain the gradients.

Weekness:
1. The proposed work is similar to the existing works on pulse-level variational quantum algorithms and quantum optimal control. [1,2,3]
2. The idea of using parameter shift to obtain gradient is not new.
3. For parameter shift, the number of circuit runs on real quantum machine is linear to the number of parameters in the circuit so when the size of the circuit is large, the number of gates will growing rapidly. For instance, the pulse length is O(2^(2n)) for quantum optimal control, so the proposed method is not very scalable either.

[1] Meitei, Oinam Romesh, et al. "Gate-free state preparation for fast variational quantum eigensolver simulations: ctrl-vqe." arXiv preprint arXiv:2008.04302 (2020).
[2] Liang, Zhiding, et al. "Variational quantum pulse learning." arXiv preprint arXiv:2203.17267 (2022).
[3] de Keijzer, Robert, Oliver Tse, and Servaas Kokkelmans. "Pulse based Variational Quantum Optimal Control for hybrid quantum computing." arXiv preprint arXiv:2202.08908 (2022).

---

> ### Author Response · Authors · 2022-08-01
> **Response Part 1**
>
> > The proposed work is similar to the existing works on pulse-level variational quantum algorithms and quantum optimal control. [1,2,3]. What are the core differences to those papers?
> > [1] Meitei, Oinam Romesh, et al. "Gate-free state preparation for fast variational quantum eigensolver simulations: ctrl-vqe." arXiv preprint arXiv:2008.04302 (2020).
> > [2] Liang, Zhiding, et al. "Variational quantum pulse learning." arXiv preprint arXiv:2203.17267 (2022).
> > [3] de Keijzer, Robert, Oliver Tse, and Servaas Kokkelmans. "Pulse based Variational Quantum Optimal Control for hybrid quantum computing." arXiv preprint arXiv:2202.08908 (2022).
>
>
> The main contribution of this paper is on connecting differentiable programming and analog quantum computing. None of the mentioned references has tackled the auto-differentiation technique for analog quantum computing.
> The contributions of our work can be summarized as follows: (a) we develop a new continuous-time parameterization (with basis functions, which is different from e.g., the parameterization in GRAPE) on analog quantum computers; (b) we design the differentiation pipeline in our framework; and (c) the correctness, effectiveness, and robustness of our framework have been demonstrated with both theoretical and empirical analysis.
>
> As will be elaborated below, none of the mentioned references [1,2,3] has contributed from the perspective of (a)(b)(c).  In particular, they didn’t discuss computing gradients analytically for analog quantum computing by using quantum machines, which our paper addressed for the first time.
>
> The first paper (Meitei et al., 2020) has been cited in our paper as Ref. [40]. This work uses GRAPE to calculate the gradient and generate the pulse. GRAPE uses classical simulation of a piecewise-constant pulse ansatz, and hence it has a different parameterization from ours.  It further suffers from *exponential* cost in classical simulation, as we mentioned in Related Works in Section 2.
>
> The second paper (Liang et al., 2022) is particularly concerned with solving classical machine learning problems with variational quantum pulses (VQP), focusing on *encoding classical training data into quantum states using pulses*. As for the pulse optimization part, they consider *gradient-free* methods such as Bayesian optimization. Hence, there is no perspective from differentiable programming in this work. Compared with our work, we develop  *gradient-based* optimization in pulse training. In addition, we further consider a wider range of applications that are not limited to classical machine learning, in contrast to (Liang et al., 2022).
>
> The third paper (de Keijzer et al., 2022) employs piece-wise constant pulses and the gradient is calculated by the existing “simultaneous perturbation stochastic approximation” (SPSA) method,  which is drastically different from our *new* Monte Carlo gradient estimation subroutine (see our algorithm 1). The gradients estimated by SPSA are much less accurate than ours, where an added comparison is in the NEW Figure 2. (NOTE:  We add more comparisons against SPSA on VQE and QAOA experiments in the revised paper in Fig. 2, as asked by reviewer FN4c.) Again, no perspective from differentiable programming in this work.  Furthermore, we briefly discuss the application of our method to superconducting qubits machines (e.g., the IBM machine), while de Keijzer et al. is particularly interested in Rydberg atoms.
>
> We hope that our differences from [1,2,3] are clear. Our major contributions do not overlap with theirs. We did cite and compare with [1] in the paper and [2,3] are very recent non-peer-reviewed papers that we are happy to incorporate into our revision. However, according to the NeurIPS 2022 FAQ, "Authors are not expected to compare to work that appeared only a month or two before the deadline."
>
> > The idea of using parameter shift to obtain gradient is not new.
>
> Chain rule was nothing new either when the neural networks were proposed, but neural networks can still be quite effective for many applications. The parameter shift rule is essentially a tool for evaluating commutators on quantum computers. We leverage it to *evaluate gradients for analog quantum computing*, while previous works focused on gradient calculation for quantum circuits. The underlying computational models are drastically different. Unlike the discrete-time circuit model, the analog model poses unique challenges for gradient evaluation because of its continuous nature. We solve this problem by the MCI technique (see our algorithm 1).
>
> We hope these explanations address your questions and we appreciate your comments. Please let us know if you have further feedback.
>
> Best, Authors

---

> > ### Author Response · Authors · 2022-08-08
> > **Discussion**
> >
> > Dear reviewer gTD7,
> >
> > We wish you had a great weekend! Thanks for reading our papers and asking questions. We have added your references to our paper. Could you please kindly let us know if there is anything we can further do or clarify that might improve your rating? Looking forward to your post-rebuttal discussion.
> >
> > Best, Authors

---

> > > ### Comment · Reviewer_gTD7 · 2022-08-10
> > > **Thanks for your response**
> > >
> > > I thank the authors for the detailed response, which solve my concerns to some extents. I increase my score accordingly. Param shift rule only applies to gates whose unitary matrix has structured eigenvalues such as +1 and -1. It would be good for the authors to explain more on the limitations of the parameter shift on the pulse level.

---

> ### Author Response · Authors · 2022-08-02
> **Response Part 2**
>
> > For parameter shift, the number of circuit runs on real quantum machine is linear to the number of parameters in the circuit so when the size of the circuit is large, the number of gates will growing rapidly. For instance, the pulse length is O(2^(2n)) for quantum optimal control, so the proposed method is not very scalable either.
>
> The high complexity of evaluating gradients for variational quantum **circuits** is precisely the motivation for our proposal of differentiable analog quantum computing using continuous-time parameterization. With abstract quantum analog machines (AQAMs) as the computational model, we detach the dependency between the number of parameters and the pulse lengths (equivalently, circuit sizes). This enables us to reduce the computational complexity to *polynomial* w.r.t. the numbers of controllable terms and sampling batch sizes, achieving far better scalability than variational circuits.
>
> Specifically in your instance, pulse length O(2^(2n)) does not imply O(2^(2n)) parameters using our parameterization: you can choose an arbitrary number of parameters with basis functions in the parameterization. On the other hand, if the runtime is polynomial in the number of parameters, our method is as scalable as classical differentiable models. The difficulty in quantum computing is that, without an appropriate auto-differentiation technique, the derivative calculation for even one parameter could require exponential numbers of simulations on classical computers, which is not scalable.
>
> > Figure 1 caption 'an' initial
>
> Thanks for your suggestion. We fixed this typo in our revision.

---

### Official Review · Reviewer_vyta · 2022-07-11

**Rating:** 8
**Confidence:** 3
**Soundness:** 4 excellent
**Presentation:** 3 good
**Contribution:** 4 excellent

**Summary:**

The paper proposes a differentiable programming framework for analog quantum computing with a specialized froward scheme based on Monte-Carlo sampling to get estimates of gradients. The rationale of the gradient estimation is justified theoretically. Moreover a sensitivity analysis to modeling errors of the quantum system is provided to handle real applications. Finally, the framework is illustrated on several examples, ranging from quantum optimization and quantum control. In all examples, the proposed framework appears not only efficient but  also outperforms previous methods by some orders of magnitudes.

**Questions:**

- One simple, yet powerful addition to the experiments would be to write down exactly the optimization problem (such as min_x f(x) with some constraints) to help the reader understand how quantum computing is used for classical problems and what are the challenges compared to classical optimization. This would greatly help other communities understand how the framework proposed by the authors differ from traditional algorithms.
- I cannot find any readme file to run the code. I appreciate that the code is commented and a simple readme/tutorial to navigate the code would also be a simple yet great improvement for the paper.

- In the definition of the AQAM: what does evolving under H_j at time t means?
- For non-experts, it could be good to either provide a quick introduction to the parameter shift rule though reference [50] is good.
- I do not understand what the authors mean by "weak" in "multi-qubit interactions are not tunable and weak compared to tunable single-qubit Halmitonians" or in "may be imprecise due to weak non-tunable terms in H_c".
- A reference for the claim "Almost on every architecture of quantum devices, the number of control signals m is at most quadratic in the number of qubits n" would be appreciated.
- For non-experts, it may be good to explain why equation (7) is presented up to Hermitian conjugate terms.
- A formal statement for equation (7) with assumptions and proof would be appreciated. Generally, a pass on the Appendix from lines 624 to 631 with detailed reasoning would greatly help clarify the theoretical contributions of the paper.
- Maybe, recall that the dagger symbol is the hermitian operator for non-experts.

**Limitations:**

The authors have properly described the limitations of their work, namely the fact that the current experiments are not very large scale although the framework had been designed to be scaled. As the authors say, the current experiments already demonstrate the potential of the framework and the "limitations" of the paper can clearly be seen as future work and not dead-ends.

**Strengths And Weaknesses:**

Strengths:
- The subject of the paper itself, namely proposing a differentiable programming framework for quantum computing is an exciting avenue for research to broaden the applicability of quantum computing.
- The approach taken by the authors, namely, considering analog quantum computing systems appears clearly driven by a good understanding of the underlying system and is original to me, though I'm not an expert in quantum computing. The authors could explain better the previous approaches. However, their experimental evaluation clearly demonstrates empirically the benefits of their approach.
- I personally come from a differentiable programming viewpoint, for which the framework posed by the authors exhibits exciting challenges, namely, computing gradient information without access to the intermediate states of the computations. The authors present a simple Monte-Carlo estimator that is easily implemented and already efficient. The framework proposed by the authors can serve as a strong baseline to build upon and provide alternative gradient estimators for analog quantum computing.
- The authors link their approach to well-known frameworks in differentiable programming such as differentiable physics which help understand the differences and challenges in this setting.
- Last but not least, the experimental evaluations are well selected and the comparisons show clearly the strength of the proposed approach compared to previous work.

Weaknesses:
- While this paper appears to be a strong contribution and an interesting bridge between differentiable programming and quantum computing, it somehow quickly passes over some technical details that could be better explained as asked int he questions section

---

> ### Author Response · Authors · 2022-08-01
> **Response**
>
> > While this paper appears to be a strong contribution and an interesting bridge between differentiable programming and quantum computing, it somehow quickly passes over some technical details that could be better explained as asked in the questions section
>
> Thanks for your suggestions. We have added more technical details to the revised paper, as indicated in blue print in the revision. Details of the changes are explained in the responses below.
>
>
> > One simple, yet powerful addition to the experiments would be to write down exactly the optimization problem (such as $min_x f(x)$ with some constraints) to help the reader understand how quantum computing is used for classical problems and what are the challenges compared to classical optimization. This would greatly help other communities understand how the framework proposed by the authors differ from traditional algorithms.
>
> We added more intuitive explanations of the problem setting connecting classical and quantum ML communities in Sec. 3.2 (p.4) of the revision.
>
> > I cannot find any readme file to run the code. I appreciate that the code is commented and a simple readme/tutorial to navigate the code would also be a simple yet great improvement for the paper.
>
> Thanks! We will add a more detailed README file and release the code on GitHub.
>
> > In the definition of the AQAM: what does evolving under $H_j$ at time t means?
>
> “Evolving under $H_j$ for time $t$” means to apply a unitary transformation $e^{-iH_j t}$, which is the time-evolution operator described by the Schrodinger equation with a constant Hamiltonian $H_j$ for time duration $t$. We improved the statement in Sec 3.2 (p.5) of the revision.
>
> > For non-experts, it could be good to either provide a quick introduction to the parameter shift rule though reference [50] is good.
>
> We added a short introduction to the parameter shift rule in Sec. 3.4 (p. 6) of the revision.
>
> > I do not understand what the authors mean by "weak" in "multi-qubit interactions are not tunable and weak compared to tunable single-qubit Halmitonians" or in "may be imprecise due to weak non-tunable terms in $H_c$".
>
> For most realistic machines, the strength of multi-qubit interactions is orders of magnitudes weaker than the single-qubit oscillation (Rabi) frequency and the driving amplitudes. Two-qubit gates are implemented by specifically designed pulses so that the driving signals and single-qubit oscillations “cancel out”, and two-qubit interactions become the major effect.
>
> > A reference for the claim "Almost on every architecture of quantum devices, the number of control signals m is at most quadratic in the number of qubits n" would be appreciated.
>
> We added more references to survey papers on different architectures in Sec. 3.5 (p. 6) of the revision.
>
> > For non-experts, it may be good to explain why equation (7) is presented up to Hermitian conjugate terms.
>
> Thanks for noting this. The presentation style in (original) Equation (7), i.e., folding the Hermitian conjugate terms, is common among physicists. We added an explanation when first using it (now in Sec. B.2 of the appendix).
>
> > A formal statement for equation (7) with assumptions and proof would be appreciated. Generally, a pass on the Appendix from lines 624 to 631 with detailed reasoning would greatly help clarify the theoretical contributions of the paper.
>
> Thanks for the suggestion. We added Lemma 3.3 in Sec. 3.6 (p. 6) of the revision, with detailed proofs in the appendix.
>
> > Maybe, recall that the dagger symbol is the hermitian operator for non-experts.
>
> We added the dagger symbol in the quantum preliminaries in Sec 3.1 (p.4) of the revision.
>
> We sincerely appreciate your comments. Please let us know if you have further feedback.
>
> Best, Authors

---

> > ### Comment · Reviewer_vyta · 2022-08-04
> > **Answer to authors**
> >
> > I thank the authors for their detailed answers to my comments and questions. I read the modifications which are appropriate. I also read the comments of Reviewer gTD7 and believe that the authors addressed them well in my opinion. Overall I maintain my score: I think this paper could inspire interesting avenues for future work (at least from a differentiable programming perspective where I come from) and the experimental evaluation is sound in my opinion (with even an additional experiment added by the authors).

---

> > > ### Author Response · Authors · 2022-08-04
> > > **Thanks for your response!**
> > >
> > > Thanks for your comments. We are also glad that our revision resolves your concerns. Please let us know whenever you have additional questions.
> > >
> > > Best, Authors

---

### Official Review · Reviewer_FN4c · 2022-07-11

**Rating:** 8
**Confidence:** 4
**Soundness:** 4 excellent
**Presentation:** 4 excellent
**Contribution:** 4 excellent

**Summary:**

This work introduces a general technique to evaluate gradients of time-evolved states, in the analog quantum computing setting.

**Questions:**

The VQE experiments compare only against finite differences, what about SPSA?

**Limitations:**

Yes

**Strengths And Weaknesses:**

This is a very interesting and conceptually (very) important work, because it allows to compute exact gradients on large analog quantum computers. While some of the technical tools have been heavily borrowed from ref 3, I believe that the extension to the analog case is crucial. Also, the numerical experiments seem to consistently indicate a superior performance to existing approaches (say, CRAB, etc).

---

> ### Author Response · Authors · 2022-08-01
> **Response**
>
> > The VQE experiments compare only against finite differences, what about SPSA?
>
> Thanks for your suggestions! We have added experiment results of SPSA on VQE and QAOA experiments in Fig. 2 of the revised paper.

---

### Author Response · Authors · 2022-08-02
**Summary**

We thank reviewers for your time reading our paper and for giving us many constructive feedbacks. We have revised the paper and the appendix to incorporate the suggestions. The modifications are highlighted in blue. We sincerely invite reviewers to take a look at our revised version. Please feel free to let us know if you have any questions or concerns.

---

### Meta-Review · Area_Chair_3gSB · 2022-08-25

**Recommendation:** Accept
**Confidence:** Certain

**Metareview:**

The paper proposes a differentiable programming framework for analog quantum computing with a specialized forward scheme based on Monte-Carlo sampling to get estimates of gradients. This idea is an exciting avenue for research to broaden the applicability of quantum computing to practical machine learning and computation. There is a clear consensus among referees that this submission constitutes an interesting and important work.  In all examples, the proposed framework appears not only efficient but also outperforms previous methods by some orders of magnitudes.

**Award:**

Yes

---

### Decision · Program_Chairs · 2022-09-14

Accept